# Father–Mother Co-Involvement in Child Maltreatment: Associations of Prior Perpetration, Parental Substance Use, Parental Medical Conditions, Inadequate Housing, and Intimate Partner Violence with Different Maltreatment Types

**DOI:** 10.3390/children10040707

**Published:** 2023-04-11

**Authors:** Joyce Y. Lee, Susan Yoon, Keunhye Park, Angelise Radney, Stacey L. Shipe, Garrett T. Pace

**Affiliations:** 1College of Social Work, The Ohio State University, Columbus, OH 43210, USA; yoon.538@osu.edu (S.Y.); radney.1@buckeyemail.osu.edu (A.R.); 2Department of Social Welfare, College of Social Sciences, Ewha Womans University, Seoul 06974, Republic of Korea; 3School of Social Work, Michigan State University, East Lansing, MI 48824, USA; kunepark@msu.edu; 4Department of Social Work, Binghamton University, Binghamton, NY 13902, USA; sshipe@binghamton.edu; 5School of Social Work, University of Nevada Las Vegas, Las Vegas, NV 89154, USA; garrett.pace@unlv.edu

**Keywords:** co-involvement in child maltreatment, fathers, mothers, family systems, risk factors, National Child Abuse and Neglect Data System

## Abstract

The current study applied a family systems approach to examine dyadic parental risk factors linked with mother–father co-involved physical abuse, neglect, sexual abuse, and emotional abuse. Parental substance use, mental health problems, disability and medical conditions, inadequate housing, economic insecurity, intimate partner violence, and prior maltreatment history were investigated as key risk factors at the dyadic parental level. Logistic regression analysis was conducted using national child welfare administrative data from the National Child Abuse and Neglect Data System. The results showed differential associations between risk factors and four child maltreatment types: physical abuse, neglect, emotional abuse, and sexual abuse. Intimate partner violence was associated with higher odds of mother–father co-involved neglect and emotional abuse. Parental substance use, inadequate housing, and prior maltreatment history were all associated with higher odds of mother–father co-involved neglect, but lower odds of physical abuse. Parental disability and medical conditions were associated with higher odds of mother-father co-involved sexual abuse, whereas parental substance use was associated with lower odds of sexual abuse. Implications include more nuanced ways of addressing multiple risk factors within the family to prevent future occurrences of child maltreatment involving both mothers and fathers.

## 1. Introduction

According to the most recently available U.S. child maltreatment data, approximately 618,000 children were abused or neglected between 1 October 2019 and 30 September 2020 [1]. This represents a national child maltreatment rate of 8.4 victims per 1000 children [1]. During the same period, there was a total of 483,285 individuals who perpetrated (i.e., caused or knowingly allowed) child maltreatment [1]. Importantly, most such individuals who perpetrated child maltreatment were parents (77.2%), with 37.6% of the cases involving mothers acting alone and 23.6% fathers acting alone. Another fifth of the cases (20.7%) involved both mothers and fathers acting together to co-perpetrate child maltreatment. This latter group has received relatively little attention in the literature despite research suggesting that child maltreatment—both physical abuse and neglect—co-perpetrated by mothers and fathers is more severe and injurious than maltreatment perpetrated by one parent alone [2].

Furthermore, while parental risk factors contributing to child maltreatment have been identified and examined at the individual level, such risk factors have not been fully considered at the dyadic parental level, especially within the context of mother–father co-involvement in child maltreatment. Using a family systems approach [3], the current study examined dyadic parental-level risk factors linked with mother–father co-involvement in different child maltreatment types (i.e., physical abuse, neglect, sexual abuse, emotional abuse) that resulted in children’s foster care entry. The focus on children entering foster care is critical given the documented vulnerabilities—including poorer mental and physical health—of the foster care population compared to children of the general population, including those from socioeconomically disadvantaged families [4]. This study makes contributions by applying family systems theory to U.S. child welfare administrative data and identifying dyadic parental risk factors linked with mother–father co-involvement of child maltreatment to inform prevention (e.g., of future recurrence) efforts.

## 2. Theoretical Framework: Family Systems Theory

Family systems theory [3,5] served as the guiding theoretical framework for the current study. The theory depicts the family as a complex social system that consists of interconnected and interdependent “subsystems” that influence each other [3,5]. One of the main tenets of family systems theory is that the family is an integrated “whole” system within which individual family members mutually influence one another [6,7]. This perspective emphasizes that the whole is greater than the sum of its parts [3]. From this perspective, simply illustrating and combining individual characteristics of each family member (or each subsystem) does not paint the full picture of family functioning.

Drawing from family systems theory, it is critical to view families as integrated whole systems and acknowledge that mothers and fathers are interrelated, interdependent subcomponents of the larger family system [3,5]. Given that mothers and fathers in the same family inherently influence one another in their thoughts and behaviors, as well as shape and are shaped by the broader contexts (e.g., the larger family system, parent–child system) within which they are embedded, merely focusing on fathers or mothers’ individual functioning would not accurately reflect the complex dynamics of families. In other words, family systems theory’s principle of “wholeness” supports the inclusion of both mothers and fathers in the data and the investigation of mother–father co-involvement in child maltreatment.

Family systems theory also applies to the field of mother–father co-involved child maltreatment in that it focuses on the interactions that occur between mothers and fathers that culminate in both members committing child abuse and neglect as joint perpetrators. From a family systems perspective, the entire family—instead of the individual mother, father, or child—is the unit of analysis, as well as the focus of intervention and treatment. Because problems such as mother–father co-involvement in child maltreatment are seen as a result of what happens in mothers’ and fathers’ interactions with one another, all behaviors—both from mothers and fathers—ought to be examined within the context of the larger family system. Furthermore, given that interactions and interdependence are emphasized within the family system, mother–father co-involvement in child maltreatment suggests alliance, communication, and coordination between mothers and fathers in co-conspiring and co-perpetrating physical abuse, neglect, emotional abuse, or sexual abuse. Mother–father co-involvement in child maltreatment also suggests the betrayal of family relations and roles, especially within the parent–child dyad. Such betrayal may leave children feeling as if they are unable to trust or comfortably interact with both their mothers and fathers [8].

## 3. Mother–Father Co-Involvement in Child Maltreatment

As noted earlier, although a substantial proportion of child abuse and neglect is committed by both mothers and fathers, little attention has been paid to mother–father co-involvement in child maltreatment. Limited research in this area suggests that mother–father co-involvement in child maltreatment is most severe and likely to lead to medical injury [2]. More specifically, using nationally representative data from the second National Survey of Child and Adolescent Well-being (NCSAW-II), Kobulsky and Wildfeuer [2] found that physical abuse and neglect committed by both mothers and fathers had the highest injury severity, as well as the largest number of co-occurring maltreatment types, when compared to those committed by either mother-alone or father-alone.

Another study using NSCAW-II data showed that child maltreatment cases were most likely to be substantiated when both mothers and fathers were involved (approximately 30% in which two parents were involved vs. 20% where either parent was involved alone) [9]. More recently, a study showed that physical abuse and sexual abuse were 1.5 times more likely to occur in families where fathers were co-involved with the mothers in committing child maltreatment compared to families where only mothers were involved [10]. That said, given data limitations, the researchers needed to combine families where fathers were solely involved in child maltreatment with families where fathers and mothers were co-involved in child maltreatment, making it difficult to distinguish which group drove the high rates of physical and sexual abuse.

Importantly, mother–father co-involvement in child maltreatment has been linked with other severe child welfare outcomes, including children’s separation from their parents and entry into foster care. For example, 10.2% of mother–father co-involved child maltreatment has been shown to result in children’s foster care entry—a number that is nearly on par with mother-only involved cases (11.8%) and substantially higher than father-only involved cases (4.2%) [9]. A focus on children entering foster care is critically needed in this area of research, given the exceptional vulnerability of this group compared to others. For example, when compared to those not placed in foster care, children placed in foster care are more likely to be in fair or poor health (4.2% vs. 3.1%), twice as likely to have learning disabilities (14.7% vs. 7.6%), three times as likely to have Attention Deficit Disorder (21.8% vs. 7.4%), six times as likely to have behavior problems (17.5% vs. 2.9%), and seven times as likely to have depression (14.2% vs. 2.0%) [4]. In addition to serious mental and physical health problems, children entering foster care most likely have previously experienced multiple incidents and types of child maltreatment [11].

The current evidence base on mother–father co-involvement in child maltreatment is quite limited, with a primary focus on physical abuse and neglect (only one study examined sexual abuse as an outcome) [9]. That is, we still know little about mother–father co-involvement in sexual abuse and emotional abuse, including factors associated with them. Furthermore, children entering foster care due to mother–father co-involvement in child maltreatment have received little attention in this area of research despite evidence documenting the multiple health challenges that they face and recent federal efforts to prevent future foster care entries [12].

## 4. Risk Factors of Mother–Father Co-Involvement in Different Child Maltreatment Types

A number of risk factors have been identified for mother–father co-involved child maltreatment, especially for physical abuse and neglect [2,9,10]. Kobulsky et al. [9] showed that the most salient risk factors of mother–father co-involvement in child maltreatment (e.g., physical abuse, neglect) included parental substance use, mental health problems, and intimate partner violence (IPV). For example, 22.3% of the cases involving mother–father co-involved child maltreatment had parental substance use as a risk factor compared to 13.4% in father-only involved and 15.9% in mother-only involved cases. Regarding IPV, mother–father co-involved cases had a substantially higher rate of IPV (17%) than those of father-only perpetration (7.4%) and mother-only perpetration (7.5%). Examining physical abuse and neglect separately, prior research has additionally identified parental mental health problems as a risk factor [2]. For example, nearly a third (32%) of the cases involving mother–father co-involved physical abuse had parental mental health problems as a risk factor compared to 19.0% in father-only involved and 27.2% in mother-only involved cases. Similar patterns were found for cases involving mother–father co-involved neglect. More broadly, the literature consistently documents low family socioeconomic resources as a risk factor for neglect [13,14].

Specific to sociodemographic characteristics, children of mother–father co-involved maltreatment cases tend to be young (between 3 and 6 years old across studies) [2,9]. Findings related to children’s sex seem mixed, with one study showing that more boys than girls tend to be victims of mother–father co-involved child maltreatment [9] and another study showing that being a girl, but not a boy, is linked with higher levels of caseworker-perceived risk of neglect cases involving both mothers and fathers [2]. Additionally, mothers’ and fathers’ race and ethnicity have shown associations with mother–father co-involvement in child maltreatment. Both mothers and fathers in co-involved maltreatment cases are more likely to be White than Black or Other races compared to parents in mother-only or father-only involved maltreatment cases [9]. Although these prior studies have not investigated mothers’ and fathers’ age in association with mother–father co-involved maltreatment, the broader child maltreatment literature suggests that younger parental age, especially younger maternal age, is a risk factor for child abuse and neglect [15,16,17].

With physical abuse and neglect being the two major outcomes of focus in the current literature, other maltreatment types (i.e., sexual abuse, emotional abuse) have not been readily examined in mother–father co-involved maltreatment research. Although Kobulsky et al. [9] was the one study that included mother–father co-involved sexual abuse, the researchers were primarily interested in examining child protective service investigation outcomes (e.g., substantiation, foster care, criminal investigation) as predicted by different perpetration configurations (e.g., mother and father co-involved, father-only involved) and child maltreatment types (i.e., physical abuse, neglect, sexual abuse). Rightly so then, dyadic parental risk factors linked with mother–father co-involved sexual abuse were not investigated. Rather, sexual abuse served as a predictor, and not a key outcome, in the researchers’ multivariate model [9].

Given limited evidence, we turn to the general child maltreatment literature to understand dyadic parental risk factors potentially linked with mother–father co-involved sexual abuse, as well as emotional abuse. A systematic review [18] found that families’ lower socioeconomic status is moderately associated with a higher risk of sexual abuse, and that parents of sexually abused children had higher rates of psychiatric symptoms. Furthermore, others have shown that maternal illness, parental substance use, and relationship conflict between parents are associated with an increased risk of child sexual abuse [19,20]. Concerning sociodemographic characteristics, the same studies found that the risk for child sexual abuse rises with child age, with 10% of the sexual abuse cases involving children aged 0 to 3, while 35.9% of such cases involved children aged 12 and older. Several studies reported consistent findings showing that girls are 2.5–3 times more likely than boys to experience sexual abuse [18,19,20,21]. With regard to race and ethnicity, one study found no significant associations between parental race and child sexual abuse [22], whereas others have found mixed evidence concerning the general effects of race and ethnicity on sexual abuse [20,23].

Specific to emotional abuse of children, a systematic review found that families’ low socioeconomic status (e.g., a household income below $15,000) is associated with a higher risk of emotional abuse [18]. Another review of the literature additionally found that poverty, parent mental illness, substance use, disability, learning difficulties, and domestic violence are linked with a higher risk of emotional abuse [24]. The same review also found that, pertaining to parental sociodemographic characteristics, early parenthood or younger parental age is associated with a higher emotional abuse risk [24]. Less is known concerning parental race and ethnicity and their associations with emotional abuse. Concerning child characteristics, prior research showed that the risk of child emotional abuse increases as the child’s age increases [25]. The evidence for child sex seems mixed, with some reporting that girls are more likely than boys to report emotional abuse [21,26] and others showing no significant links between child sex and emotional abuse [27,28].

In summary, despite the documented prevalence and severe consequences of mother–father co-involvement in child maltreatment, little attention has been paid to this topic. Further, not only are there very few studies in this area, but also available studies have been limited in terms of data (e.g., combination of disparate groups such as mother–father co-involved and father-only involved maltreatment cases; use of small and local samples). As a case in point, only two studies that we know have used national data, with a primary focus on physical abuse and neglect and their risk factors [2,9]. Mothers’ and fathers’ co-involvement in other types of child maltreatment, including sexual abuse and emotional abuse, have not been fully considered, along with related dyadic parental risk factors.

## 5. The Current Study

Using national child welfare administrative data, the current study aimed to apply a family system approach examining risk factors at the dyadic parental level associated with different types of substantiated child maltreatment (i.e., physical abuse, neglect, sexual abuse, emotional abuse) involving both mothers and fathers, leading to children’s eventual foster care entry. That is, the current study’s sample likely represents some of the most severe child maltreatment cases in the U.S. child welfare system. Informed by family systems theory and prior research in this area, we primarily hypothesized that risk factors, including parental substance use, mental health problems, medical conditions, financial and material hardship (e.g., inadequate housing, economic insecurity), and IPV would be associated higher risks of mother–father co-involved physical abuse, neglect, sexual abuse, and emotional abuse, controlling for parental and child characteristics. The current study makes key contributions to the literature, including advancing the knowledge in an underexamined area in child maltreatment research, applying family systems theory in analyzing national child welfare administrative data, and identifying risk factors linked with mother–father co-involvement in different child maltreatment types to inform programs and policies aimed at maltreatment prevention.

## 6. Materials and Methods

### Data

Data for the current study came from the National Child Abuse and Neglect Data System (NCANDS), which is a federally sponsored project of the Children’s Bureau at the U.S. Department of Health and Human Services. All states that receive federal funds provide data on children who have been maltreated per the Child Abuse Prevention and Treatment Act of 1996 [29]. Fifty states, the District of Columbia, and Puerto Rico submit two files—NCANDS Agency File with aggregate-level data and NCANDS Child File with case-level data—to the Children’s Bureau each year. Specific to the NCANDS Child File, states’ case-level data involve child-specific records for each report of alleged child abuse and neglect that received a child protective service response and resulted in a disposition during the reporting year [29].

The NCANDS Child File is available from the National Data Archive on Child Abuse and Neglect (NDACAN) in one-year batches, based on the federal fiscal year of the date of the report disposition. Federal fiscal years run from October 1 through September 30. Within a fiscal year of data, the NCANDS Child File has one row per unique Report–Child pair. A ReportID may repeat if there is more than one child on the report, and a ChildID may repeat if the child appears on more than one report. We opted to use the report date rather than the disposition date, because disposition dates may be drawn out over an unpredictable period for various reasons, and because we were interested in cases where a child was removed from the home, and our criteria for the study outcomes called for cases where the removal date was on or after the report date.

The current study focused on entire child maltreatment reports in the United States pertaining to FY 2018. To construct a pool of cases where the report date was in FY2018, we first pooled three years of the Child File (FY2018–2020) to identify eligible cases where the report date was in FY2018, but the case was disposed in a later year. From this pool, we found 12,385,182 Report–Child pairs, of which 4,339,283 had a report date in FY 2018. Of the 4,339,283 cases, there were 729,854 cases where a child was a victim of substantiated or indicated child maltreatment. It was important to focus on these cases because perpetration data were only available for those cases where child maltreatment was substantiated or indicated [29].

## 7. Analytic Sample

Participants for the current study included families in which biological mother–father pairs were identified as co-perpetrating maltreatment against a child and that maltreatment subsequently led to foster care entry of the same child. Although non-biological caregivers also engage in child maltreatment [1], the current study focused on biological mothers and fathers of the same child since research shows that more biological parents than non-biological parents perpetrate child maltreatment (e.g., [30,31]), as well as given complexities with the NCANDS data structure. Of the 729,854 relevant cases in the original NCANDS Child File dataset, there were 145,201 cases in which children were removed from their homes and entered foster care on or after the child maltreatment report date. Of these, 26,195 cases had co-perpetrators, namely, the biological father and biological mother, who were both responsible for the caregiving of the focal child and who were involved with or knowingly allowed child maltreatment to occur to the same child.

We kept all relevant cases regardless of the number of maltreatment reports present within the same family. For families with only one child maltreatment report, cases were selected if both the biological father and mother were identified as perpetrators of the same maltreatment type. For families with more than one child maltreatment report, cases were selected if both the biological father and biological mother were concurrently substantiated for at least one maltreatment type. For cases with multiple children under the same report, we retained cases with the youngest child and were thus left with 16,971 cases. This was based on statistics showing that young children have some of the highest child maltreatment victimization rates (e.g., victimization rate of 25.1 per 1000 children for those younger than 1 year old vs. 7.1 per 1000 for children for those aged 10 years) [1]. Of these, 578 duplicate perpetrator and child IDs were removed, resulting in 16,393 cases.

Regarding risk factors, because not all states collect information on parental risk factors associated with child maltreatment, we first conducted a detailed check of missing data patterns in caregiver risk factors across states, as recommended by professionals managing the NDACAN data (e.g., Michael Dineen, personal communication, 1 July 2022). Of the 42 states available in the FY 2018 NCANDS Child File, 11 states provided complete data on parental risk factors. These states included Arkansas, Delaware, Florida, Indiana, Michigan, Minnesota, Mississippi, Nebraska, South Dakota, Texas, and Utah. Based on this data inspection and consultations from NCANDS, only those cases from 11 states providing complete risk factor data were retained, which involved dropping 9191 cases from the remaining states. Finally, we dropped 126 fathers and 80 mothers whose race and ethnicity could not be identified. The final analytic sample contained 6996 unique families, with a triad involving a co-perpetrating mother, a co-perpetrating father, and a victimized child being the unit of analysis.

## 8. Measures

### 8.1. Independent Variables

Seven parental risk factors served as independent variables. These parental risk factors indicated whether a relevant risk was present in a family for at least one of the parents. Since information on who specifically in the family had a particular parental risk factor was not provided in the NCANDS data, the parental risk factors measured in this study represent risk at the dyadic parental level, indicating that at least one of the parents has the risk factor.

*Substance Use*. Parental substance use was captured using two binary variables, including parental alcohol use (yes/no) and drug use (yes/no). NCANDS defined parental alcohol use and drug use as compulsive use of alcohol and drugs, respectively, that is not of temporary nature [29]. These two variables were combined and then converted into a binary substance use variable, indicating whether a risk of parental alcohol use and/or drug use was present in the family (yes/no).

*Mental Health Problems*. Parental mental health problems were coded as a binary variable (yes/no) defined within NCANDS as a clinically diagnosed condition displaying at least one of the following characteristics over a long period and to a marked degree: (a) inability to build or maintain satisfactory interpersonal relationships; (b) inappropriate types of behaviors or feelings under normal circumstances; (c) general pervasive mood of unhappiness or depression; (d) tendency to develop physical symptoms or fears linked with personal problems. Mental health problems were diagnosed based on the Diagnostic and Statistical Manual of Mental Disorders (DSM) [29].

*Disability and Medical Conditions*. Parental disability and medical conditions were captured using five binary variables, including intellectual disability, learning disability, physical disability, visual or hearing impairment, and other medical conditions. NCANDS provided definitions for each of these conditions [29]. Intellectual disability is defined as a clinically diagnosed condition involving significantly less than average intellectual functioning that exists concurrently with limits in adaptive behaviors that negatively affect socialization and learning. Learning disability is defined as a clinically diagnosed disorder (e.g., perceptual disability, brain injury, dyslexia, developmental aphasia) in basic psychological processes involved in understanding or using language (spoken or written) that may display itself in a limited ability to listen, think, speak, and write. Physical disability is defined as a clinically diagnosed condition (e.g., cerebral palsy, spina bifida, multiple sclerosis) that negatively affects daily motor functioning. Visual or hearing impairment is a clinically diagnosed condition related to a visual impairment or permanent or fluctuating hearing or speech impairment that significantly affects functioning. Other medical conditions is defined as medical conditions (e.g., chronic illnesses, HIV/AIDS) other than those mentioned above that significantly impact functioning or require special medical care [29]. These five variables were combined and then converted into a binary variable (yes/no), indicating whether a risk of disability and medical conditions was present at the dyadic parental level.

*Intimate Partner Violence*. IPV was a binary variable (yes/no) defined within NCANDS as any abusive, violent, coercive, forceful, or threatening act or word inflicted by one parent on another [29].

*Inadequate Housing*. Inadequate housing was a binary variable (yes/no) defined within NCANDS as having housing facilities that are substandard, overcrowded, unsafe, or otherwise inadequate for living with the child (e.g., homelessness) [29].

*Economic Insecurity*. Economic insecurity was captured using two binary variables (yes/no), including financial problems and receipt of public assistance. According to NCANDS, financial problems were defined as an inability to provide sufficient financial resources to meet the minimum needs of the family. The receipt of public assistance served as a proxy variable for families’ poverty status and was defined as participation in social service programs, including, but not limited to, Temporary Assistance for Needy Families (TANF), General Assistance, Medicaid, Supplemental Security Income (SSI), Supplemental Nutrition Assistance Program (SNAP), and Special Supplemental Nutrition Program for Women, Infants, and Children (WIC) [29]. These two variables were combined and then converted into a binary variable (yes/no), indicating whether a risk of economic insecurity was present.

*Prior Perpetration of Child Maltreatment*. The presence of a parent who was previously involved with child maltreatment was captured using two binary variables (yes/no)—one for mothers and another for fathers. According to NCANDS, a prior abuser or perpetrator is defined as an individual who caused or knowingly allowed child maltreatment to occur with a previous determination (i.e., occurred before the current child maltreatment report disposition date) in the state’s information system of substantiated or indicated child maltreatment [29]. Because other parental risk factors in this study represent a risk at the dyadic parental level, the prior perpetrator variables for mothers and fathers were combined and then converted into a binary variable (yes/no), indicating the presence of a parent(s) who had previously perpetrated child maltreatment in the family.

### 8.2. Dependent Variables

Four types of child maltreatment served as individual dependent variables: (1) physical abuse; (2) neglect; (3) sexual abuse; (4) emotional abuse. NCANDS includes several categorical variables reflecting various types of child maltreatment as defined by policy and state law. Given missing data on some of the maltreatment types (e.g., medical neglect, sex trafficking, other abuse), we primarily focused on the aforementioned four types of child maltreatment. Binary variables (yes/no) were created for each of the four child maltreatment types.

### 8.3. Sociodemographic Control Variables

As informed by prior research in this area, we used several sociodemographic variables of children and parents to be included as control variables in our models. Regarding child variables, we included a continuous variable for child age and a binary child sex variable (0 = male; 1 = female). For parents, both mothers’ and fathers’ ages were included as continuous variables. For mothers’ and fathers’ race and ethnicity, caseworkers asked and subsequently entered relevant information into administrative records. We coded couples’ race and ethnicity as four mutually exclusive binary variables (yes/no): Non-Hispanic/Latinx Black, Non-Hispanic/Latinx White, Hispanic/Latinx, and Other (i.e., Native Hawaiian or Pacific Islander [NHPI], Asian, American Indian or Alaska Native [AIAN], Multiracial or Interracial). In our models, Non-Hispanic/Latinx White race served as the reference category and thus three separate binary variables for race and ethnicity—Hispanic/Latinx, Non-Hispanic/Latinx Black, and Other—were included in the models.

Additionally, given the wide variations in definitions and practices in the investigation and substantiation of child maltreatment across states [32,33,34], we created dummy variables to control for state variation for the following 11 states reflected in our analytic sample: Arkansas, Delaware, Florida, Indiana, Michigan, Minnesota, Mississippi, Nebraska, South Dakoda, Texas, and Utah. Using alphabetical order, Arkansas served as the reference category and thus 10 states were included as separate dummy variables in our models.

## 9. Analysis Plan

Descriptive analysis was conducted by obtaining the means and standard deviations of the study variables. Next, bivariate analysis was conducted by examining the correlations between study variables. Multicollinearity was tested using the variance inflation factor (VIF), with VIF values of <3 indicating low correlations between variables [35]. These analyses were conducted using Stata 17 [36].

For our main analyses, we used logistic regression analyses conducted in Mplus 8 [37]. Specifically, we ran four separate logistic regression models to examine the associations between risk factors and four different maltreatment types (i.e., physical abuse, neglect, sexual abuse, and emotional abuse) that were co-perpetrated by mothers and fathers in the same family. All four models controlled for child sex, child age, perpetrating father’s age, perpetrating mother’s age, couple’s race and ethnicity, and state variations in definitions of child maltreatment. Maximum likelihood estimation with robust standard errors was used.

Missing data in this study ranged from 0% to 28% (e.g., disability and medical conditions variable). Tests of missing data mechanisms suggested that data were either missing completely at random (MCAR) where the probability of data missing is random and not dependent on any values in the data, or missing at random (MAR) where the probability of data missing is related to some of the observed data. Logistic regression was used to test missing data mechanisms, and our results showed that most variables with missing data were MAR, with missing values being significantly associated with other study variables. The exceptions were parental substance use, mother’s age, and father’s age, which were MCAR such that logistic regression results showed no associations between missing values on these variables and other study variables. Missing data were handled via full information maximum likelihood (FIML), which assumes that data are at least MAR and estimates parameters by using all available data.

## 10. Results

### 10.1. Preliminary Results

The vast majority of families in this study (over 73%) had only one type of substantiated or indicated child maltreatment, in which both the biological father and mother were co-involved. The remaining families with more than one type of maltreatment, ranging between two to four different types, involved biological fathers and mothers, who were both substantiated for at least one of the child maltreatment types. Descriptive statistics of study variables and their correlations are presented in Table 1.

Briefly, of the different mother–father co-involved child maltreatment types, approximately 93% of the cases were neglect, followed by 14% physical abuse, 2% sexual abuse, and 1% emotional abuse. With respect to risk factors, 57% of the cases had at least one parent in the family who had previously perpetrated child maltreatment, while 43% of the cases involved parental substance use. Approximately 8% of the cases involved both parental mental health problems, as well as disability and medical conditions. A fifth of the families experienced inadequate housing, and over half experienced economic insecurity. A third of the cases involved IPV in the family. On average, mothers were 30 years old and fathers were 34 years old. Over half of the couples were Non-Hispanic/Latinx White (58%), followed by Other race (18%), Non-Hispanic/Latinx Black (14%), and Hispanic/Latinx (9%). Children were on average 3 years old.

The correlation results showed that all the risk factors measured in this study were significantly correlated with at least one of the child maltreatment outcomes. Similar patterns were found for the sociodemographic control variables. Based on these correlation tests, all variables noted earlier were included in the main analyses. No multicollinearity was found between study variables, with the main independent variables having VIF values of <2.

### 10.2. Multivariable Logistic Regression Results

The multivariable logistic regression results for each child maltreatment outcome are shown in Table 2. Specific to the model examining physical abuse as the outcome, logistic regression results showed that the presence of a prior perpetrator in the family was associated with 0.50 times lower odds of mother–father co-involved physical abuse (AOR = 0.50, 95% CI [0.42,0.61]). Parental substance use (AOR = 0.83, 95% CI [0.69,0.99]) and inadequate housing (AOR = 0.63, 95% CI [0.48,0.81]) were associated with 0.17 times and 0.37 times lower odds of mother–father co-involved physical abuse, respectively. Furthermore, each additional year in child age was associated with 0.05 lower odds of mother–father co-involved physical abuse (AOR = 0.95, 95% CI [0.92,0.98]). On the contrary, Non-Hispanic/Latinx Black couple race was associated with 1.41 times higher odds of mother–father co-involved physical abuse compared to Non-Hispanic/Latinx White couple race (AOR = 1.41, 95% CI [1.12,1.78]).

With regards to the model examining neglect as the outcome, the results showed that parental substance use was associated with 2.72 times higher odds of mother–father co-involved neglect (AOR = 2.72, 95% CI [1.82,4.06]). Similarly, having a prior perpetrator in the family (AOR = 2.26, 95% CI [1.57,3.24]) and inadequate housing (AOR = 2.29, 95% CI [1.44,3.64]) were associated with 2.26 times and 2.29 times higher odds of mother–father co-involved neglect, respectively. Furthermore, IPV was associated with 1.93 times higher odds of mother–father co-involved neglect (AOR = 1.93, 95% CI [1.34,2.78]). Non-Hispanic/Latinx Black couple race was associated with 0.40 times lower odds of mother–father co-involved neglect (AOR = 0.60, 95% CI [0.39,0.92]).

Concerning the model examining sexual abuse as the outcome, our findings showed that parental disability and medical conditions were linked with 1.97 times higher odds of mother–father co-involved sexual abuse (AOR = 1.97, 95% CI [1.03,3.75]). Older child age (AOR = 1.21, 95% CI [1.15,1.27]) and female child sex (AOR = 4.24, 95% CI [2.56,7.02]) were linked with 1.21 times and 4.24 times higher odds of mother–father co-involved sexual abuse, respectively. Parental substance use was linked with 0.81 times lower odds of mother–father co-involved sexual abuse (AOR = 0.19, 95% CI [0.10,0.36]). Moreover, Non-Hispanic/Latinx Black couple race was linked with 0.59 times lower odds of mother–father co-involved sexual abuse compared to Non-Hispanic/Latinx White couple race (AOR = 0.41, 95% CI [0.19,0.87]).

Specific to the model examining emotional abuse as the outcome, the results showed that IPV was associated with 6.71 times higher odds of mother–father co-involved emotional abuse (AOR = 6.71, 95% CI [3.64,12.37]). Furthermore, Other couple race (e.g., AIAN, NHPI, Asian, Multiracial, Interracial) (AOR = 2.11, 95% CI [1.17,3.81]) and older child age (AOR = 1.08, 95% CI [1.01,1.15]) were associated with 2.11 times and 1.08 times higher odds of mother–father co-involved emotional abuse, respectively.

As part of our sensitivity analysis, we also conducted subanalyses. Specifically, we examined our main models by child sex (female vs. male) and child developmental stage: infancy and toddlerhood (0–2 years), early childhood (3–5 years), school age (6–12 years), and adolescence (13–17 years). Both similarities and differences were found across child sex and developmental stage, which are detailed in Appendix A. 

## 11. Discussion

Applying a family systems approach [3,7] and utilizing national child welfare administrative data, we sought to examine dyadic parental-level risk factors associated with different types of child maltreatment co-perpetrated by mothers and fathers. This study contributes to the limited literature on mother–father co-involvement in child maltreatment, especially by employing the family systems theory. Family systems theory argues for viewing families as integrated whole systems, with mothers and fathers serving as interrelated and interdependent subsystems of larger family systems. Not only are mothers’ and fathers’ data used jointly, but also the entire family serves as the unit of analysis in family systems-informed studies such as ours. Given the generally atheoretical nature of prior studies on the topic of mother–father co-involvement in child maltreatment [2,9,10], the application of family systems theory is a key contribution of this study, challenging current theoretical models on child abuse to consider the joint roles of mothers and fathers and risk factors that occur at the parental dyad or family level. Our study also importantly informs and contributes to maltreatment prevention policy and practice. All the examined dyadic parental risk factors, except for mental health problems and economic insecurity, were significantly associated with at least one of the child maltreatment outcomes, even after controlling for sociodemographic factors, supporting our main hypothesis. In particular, prior perpetrator in the family, substance use, and IPV were found to be salient predictors of child maltreatment committed by both mothers and fathers, though the strengths and directions of the associations varied depending on the type of maltreatment—a finding that seems to support links between distinct parental risk factors and maltreatment types in the broader literature [38].

For physical abuse, we found that having a prior perpetrator in the family, substance use amongst parents, inadequate housing, and older child age were associated with lower odds of mother–father co-involved physical abuse. These findings contradict the findings of prior research that indicated a prior history of maltreatment, parental substance use, and housing instability as key risk factors for child physical abuse, including mother–father co-involved physical abuse [2,9,39,40,41]. It is possible that challenges associated with severe substance use and housing insecurity make it difficult for mothers and fathers to engage in parenting in general, let alone behaviors indicative of physical abuse of their children. That is, parents who are intoxicated due to alcohol or drugs, as well as those who are struggling to find adequate housing for their families, may not have sufficient bandwidth for any type of parenting, even poor or harsh parenting. Regarding the inverse association between a prior perpetrator in the family and child physical abuse, it may be that additional resources and services (e.g., positive parenting education programs) provided to parents with prior child welfare involvement, coupled with increased scrutiny and ongoing monitoring of the families (e.g., informal social control), contribute to a lower likelihood of (recurrence of) physical abuse [42].

With regards to neglect as the outcome, the results revealed that the presence of a prior perpetrator in the family, substance use amongst parents, inadequate housing, and IPV are key risk factors associated with higher risks of mother–father co-perpetrated neglect. For the most part, these findings support those from prior research [9,13,14]. Moreover, they are interesting in that a prior perpetrator in the family, parental substance use, and inadequate housing were all associated with lower odds of physical abuse but with higher odds of neglect. There is evidence to suggest, for example, that parental substance use may have a differential impact on child abuse and neglect based on different patterns of substance use and different maltreatment types [40]. Together, these results suggest that the direction of the relationships between various dyadic parental characteristics and child maltreatment can vary by the type or nature of maltreatment—acts of omission (e.g., neglect) versus acts of commission (e.g., physical abuse). Additionally, exposure to IPV was identified as a risk factor of mother–father co-perpetrated neglect, affirming prior work that indicated a positive link between IPV and child neglect [43,44]. Previous studies have suggested that fathers who perpetrate IPV tend to be authoritarian, less involved with their children, neglectful, and verbally abusive towards their children [45,46,47]. At the same time, mothers who are victims of IPV are at risk of experiencing elevated levels of depression and parenting stress, which can contribute to child neglect [48,49,50].

When sexual abuse was examined as the outcome, disability and medical conditions amongst parents were linked with higher odds of mother–father co-involved sexual abuse, while parental substance use was linked with lower odds of sexual abuse. Studies have indicated parental (especially maternal) illness and disability as risk factors for child sexual abuse [19,51]. Role reversal and parentification resulting from mothers’ illness and disability might create sexually abusive family systems with enmeshed and permeable boundaries where parents sexually abuse children for their sexual needs [52]. As for the unexpected finding of an inverse relationship between parental substance use and sexual abuse, similar to our speculation about the physical abuse findings, it might be that parents who are intoxicated are generally absent and uninvolved in parenting and child-rearing. However, more research is needed to disentangle the complex relationship between parental substance use and childhood physical and sexual abuse.

Finally, for emotional abuse, IPV was highlighted as a salient risk factor, while none of the other risk factors examined in this study had statistically significant associations with mother–father co-involved emotional abuse. A positive link between IPV and emotional abuse is in accordance with prior findings that have shown co-occurrence between IPV and child emotional abuse [43,44,49,53]. It can be said that all children exposed to IPV and family conflict, by virtue of their adverse and traumatic experiences in the home, are also experiencing psychological or emotional abuse [54]. Further, drawing from family systems theory [6,7], IPV between parents (i.e., marital or couple system) can have harmful spillover effects and negatively affect children’s emotions and psychological states via the parent–child system.

Notably, economic insecurity and mental health problems did not emerge as risk factors for any of the maltreatment types. In other words, when controlling for other risk factors and sociodemographic characteristics, economic insecurity and parental mental health problems were not predictive of child maltreatment types above and beyond the effects of other risk factors. These findings are inconsistent with previous findings that suggest positive associations of economic insecurity and parental mental health problems with child maltreatment [55,56,57]. The null findings in the current study might be due to the measures used to assess parental mental health problems and economic insecurity. For example, the operationalization of mental health problems in our study was limited to DSM-based, clinically diagnosed mental health conditions and we did not consider undiagnosed mental health symptoms that might be related to an increased risk of child maltreatment.

For economic insecurity, we used a binary (yes/no) variable versus a comprehensive financial or material hardship index. That is, we were limited in capturing the severity or level of material hardship, which has been shown to be one of the most robust predictors of maltreatment in a systematic review of the link between economic insecurity and child maltreatment [55]. Two variables—financial problems and receipt of public assistance—were used in creating our binary economic insecurity variable. In checking that the combination of these two variables did not wash out any potential effects of economic insecurity, we conducted post-hoc analyses. Specifically, we disaggregated the economic insecurity variable and included financial problems and receipt of public assistance as separate predictors in our models. The post-hoc analyses did not yield any significant results, suggesting general limitations of the economic-related measures in the data (e.g., not robust enough to capture economic conditions of families).

Finally, although race and ethnicity were not examined as focal predictors of maltreatment in the current study, the study results revealed interesting findings regarding the link between race and maltreatment types. Non-Hispanic/Latinx Black couple race was associated with higher odds of mother–father co-involved physical abuse but was associated with lower odds of other maltreatment types, such as neglect and sexual abuse. These findings might reflect the lack of a culturally sensitive lens and the existence of negative bias toward Black mothers’ and fathers’ parenting, especially in the context of child discipline, in current child welfare practice. That is, the use of physical discipline, which is considered normative practice in the Black community, may be misinterpreted as physical abuse for Black parents [58,59]. Further, previous studies have suggested that Black families are at higher risk for being reported to child protective services and substantiated for physical abuse compared to White families [60,61]. Child welfare workers may need to adopt a more culturally specific and responsive lens in understanding parenting among Black families.

## 12. Limitations and Future Research Directions

The current study has a number of limitations. Given the cross-sectional nature of the NCANDS data, no causal conclusions can be drawn from the results. Future research could use longitudinal data to further examine the relations between parental risk factors and mother–father co-involvement in child maltreatment over time.

Although we limited our analytic sample to biological mothers and fathers only, non-biological caregivers engage in child maltreatment and thus it would be worthwhile in the future to examine similarities and differences across different co-perpetrating groups (e.g., biological parents only, non-biological parent and biological parent, non-biological parents only). This also extends to an examination of how siblings of the victimized child are impacted (e.g., whether equally abused and what types of maltreatment).

Given data limitations, including the fact that we could not determinewhich parental risk factors belonged to whom (i.e., fathers or mothers), we were not able to examine specific maternal and paternal risk factors contributing to co-involvement in child maltreatment. Future research using data that specify these risk factors by maternal and paternal status would allow for more nuanced findings that help identify who might be carrying an increased risk of putting the mother–father dyad at risk of co-perpetrating child maltreatment. We were also limited to the measures in NCANDS, which were collected primarily for administration and program improvement purposes and not research. As such, measures such as mothers’ and fathers’ income and educational levels, as well as other material hardships (e.g., food insecurity, access to utilities), were missing. We might have seen expected results concerning the documented effects of economic insecurity on child maltreatment if we had, and used, for example, parents’ income information. Additional research examining the role of financial and material hardship on mother–father co-involvement in child maltreatment is needed.

Relatedly, given data limitations, we were unable to include measures that capture social context as a potential risk factor. For example, future research would do well to examine the total number of children in the household and how far family members, such as grandparents and other relatives, live from the target families as social context (e.g., crowded house, lack of accessible social and family support) risk factors associated with mother–father co-involvement in child maltreatment. Furthermore, the absence of relevant information in the data prevented us from including additional characteristics of the children and parents (e.g., parental employment status, family composition, participation in parenting education programs, children’s disability status) as potential risk factors in our analytic models. Overall, future research in this area should aim to include and examine such social context and family characteristics in their associations with mother–father co-involved child maltreatment.

A future application of family systems theory in this area of research would be to examine how mothers and fathers establish alliances, communicate, and execute child maltreatment jointly. Because of limited data availability, we were unable to examine such dyadic and interpersonal processes that culminate in mother–father co-involvement in child maltreatment. Furthermore, although not feasible for the current study, future research could extend the family systems theory to examine the consequences of mother–father co-involvement in child maltreatment on parent–child dyads (e.g., betrayal, loss of family ties, mistrust children may feel towards their parents), as well as child wellbeing outcomes (e.g., emotional insecurity, internalizing problem, externalizing problems).

Sample sizes for some of the models, especially those of emotional abuse and sexual abuse, were small which warrants caution in interpreting their results. Future studies with a larger number of cases are needed to replicate our study results concerning emotional and sexual abuse.

Finally, our results are not nationally representative as the current study’s analysis focused on families from only 11 states that consistently collected parental risk factor data submitted to the Children’s Bureau. Although our results may be generalizable to the 11 states, we caution the readers in doing so, as states in NCANDS do not collect data in a uniform manner. States have different policies and procedures for collecting their child welfare data, including parental risk factors. Future research may consider using databases such as the State Child Abuse and Neglect (SCAN) policies database, which compiles state definitions and policies related to child maltreatment and risk factors, to identify state variations in parental risk factor assessment and reporting in relation to mother–father co-involved child maltreatment. Standardized efforts may be needed in the future to ensure that the majority of states are collecting information on the same indicators. This would allow for a larger number of states, if not all states, in NCANDS to be used in future research that aims to elucidate the role of factors in different mother–father co-involved child maltreatment types. Moreover, although our study was unable to provide direct evidence, state characteristics (e.g., how strong their child protection laws are, political affiliation, amounts of rural areas and poverty, and overall levels of childhood trauma) are likely linked with the prevalence of mother–father co-involvement and future research would do well to examine such state characteristics.

## 13. Implications for Practice and Policy

Around one-in-five cases of substantiated maltreatment are attributed to both the child’s mother and father [1]. Practitioners and policymakers may find success in preventing mother–father co-involvement in child maltreatment by addressing specific risk factors in the family system. Our results suggest that addressing the presence of a prior perpetrator in the family, parental substance use, disability and medical conditions, inadequate housing, and IPV are possible levers for programmatic and policy change in the prevention of future mother-father co-involved child maltreatment incidents (e.g., recurrence). However, it is important to be mindful of how efforts to reduce one type of maltreatment may impact those to decrease another type of maltreatment. For example, interventions that aim to prevent IPV may also help prevent emotional abuse and neglect, though perhaps not physical or sexual abuse. Similarly, helping families gain adequate housing may help prevent neglectful parenting, but may also be linked to more situations in which parents physically punish their children, which may escalate to physical abuse. Child welfare practitioners would do well to engage in comprehensive assessments of risk factors at the dyadic parental and family levels, as well as engage in service referrals, resource provisions, and treatments that ensure the well-being of families and their children across multiple domains (e.g., housing, physical and mental health, interparental and parent–child relationships, criminal justice involvement).

Risks for mother–father co-involvement in certain types of maltreatment varied by couples’ race and ethnicity. For example, Black mothers and fathers were at a greater risk than White families of being co-involved in perpetrating physical abuse while also being at a lower risk than White families of being co-involved in perpetrating neglect and sexual abuse. Given the importance of preserving family ties and eliminating oppressive practices, including unnecessary separation of children from their parents and families, it would be critical for child welfare practitioners to apply a culturally sensitive and anti-racist lens to reporting, investigating, and substantiating child maltreatment [62,63]. They should be aware of potential biases and cultural insensitivity that may lead to disparate child welfare outcomes across racial and ethnic categories, as well as employ culturally responsive services. For example, family group decision making (FGDM) is a child welfare practice that reflects the cultural values of kinship and community often seen in Black and Indigenous groups [63]. FGDM brings together both immediate and extended family members, trained facilitators, community members, and agency personnel to develop child safety and well-being plans [63]. Through a communal process, family members are encouraged to define their own groups, have a voice in the decision-making process, and remind other stakeholders that children belong with their families and within their kinship networks [63].

Furthermore, child maltreatment prevention programs and social services should take into account the cultural practices of child discipline and upbringing (i.e., physical punishment) in different racial and ethnic communities. Such programs and services should not only recognize overlaps between physical punishment and some forms of child maltreatment such as physical abuse, but also aim to understand where physical punishment might be stemming from for parents of color, including Black parents (e.g., historical injustices such as slavery, fear that their children will get into trouble with law enforcement, high levels of socioeconomic disadvantage, and ensuing economic stress). At a more macro level, policymakers should aim to change social norms around issues such as physical punishment to support caregivers in their efforts to raise their children [64].

## Figures and Tables

**Table 1 children-10-00707-t001:** Descriptive Statistics and Correlations Between Study Variables (*N* = 6996).

Variable	1	2	3	4	5	6	7	8	9	10	11	12	13	14	15	16	17	18	19
1. Physical abuse	—																		
2. Neglect	0.21 ***	—																	
3. Sexual abuse	−0.04 **	0.00	—																
4. Emotional abuse	0.10 ***	0.10 ***	0.03 **	—															
5. Prior perpetration	0.08 ***	0.03 **	−0.03 *	−0.01	—														
6. Substance use	0.10 ***	0.12 ***	0.10 ***	0.05 ***	0.10 ***	—													
7. Mental health problems	0.06 ***	0.02	0.05 ***	−0.01	−0.01	0.10 ***	—												
8. Disability and medical conditions	−0.04 **	0.01	0.04 **	−0.02	−0.01	0.02	0.19 ***	—											
9. Inadequate housing	0.06 ***	0.00	−0.04 **	0.02	−0.04 **	0.06 ***	0.05 **	0.05 ***	—										
10. Economic insecurity	0.01	0.02	−0.04 *	−0.03	0.23 ***	0.18 ***	0.12 ***	0.04 **	0.15 ***	—									
11. Intimate partner violence	−0.03 **	−0.03 **	0.06 ***	0.07 ***	−0.03 *	0.06 ***	0.04 **	0.00	0.08 ***	−0.04 **	—								
12. Child age	0.05 ***	0.02	0.21 ***	0.08 ***	0.02	0.09 ***	0.01	0.00	−0.02	−0.01	0.08 ***	—							
13. Child sex (girl)	0.00	−0.01	0.10 ***	0.00	−0.01	−0.01	0.02	−0.01	0.01	0.00	0.00	0.06 ***	—						
14. Father’s age	0.08 ***	0.04 ***	0.09 ***	0.05 ***	0.11 ***	0.03 *	0.06 ***	0.04 **	0.04 **	0.06 ***	0.08 ***	0.44 ***	0.04 ***	—					
15. Mother’s age	0.08 ***	0.03 **	0.11 ***	0.07 ***	0.07 ***	0.00	0.04 **	0.02	0.02	0.02	0.05 ***	0.57 ***	0.03 **	0.69 ***	—				
16. White	−0.04 **	0.09 ***	0.00	−0.01	0.03 *	0.07 ***	0.00	0.02	0.04 **	0.07 ***	0.08 ***	0.05 ***	0.00	0.06 ***	0.08 **	—			
17. Latinx	0.01	−0.02	0.03 *	0.00	0.05 ***	−0.02	−0.03 *	−0.01	0.05 ***	0.12 ***	0.05 ***	0.01	0.02	−0.05	−0.02	0.37 ***	—		
18. Black	0.03 **	0.10 ***	−0.02	−0.02	0.03 *	0.11 ***	−0.01	−0.01	−0.02	0.02	0.03 *	0.00	−0.01	−0.02	0.05 ***	0.48 ***	0.13 ***	—	
19. Other	0.01	−0.02	0.00	0.02 *	−0.03 *	0.03*	0.04 *	0.00	0.00	−0.02	0.04 ***	0.07 ***	0.00	−0.02	0.04 ***	0.56 ***	0.15 ***	0.19 ***	—
*M* or %	14.28	92.94	2.00	1.24	57.20	43.22	8.42	7.86	19.97	52.04	33.27	2.96	48.21	33.79	30.14	58.38	9.02	14.29	18.31
*SD*	—	—	—	—	—	—	—	—	—	—	—	4.14	—	8.89	7.07	—	—	—	—

*Notes*. * *p* < 0.05. ** *p* < 0.01. *** *p* < 0.001. M = Mean. SD = Standard Deviation.

**Table 2 children-10-00707-t002:** Results of Multivariable Logistic Regression Models Predicting Different Child Maltreatment Types.

Variables	Physical Abuse		Neglect		Sexual Abuse		Emotional Abuse	
	AOR 95% CI	*p*	AOR 95% CI	*p*	AOR 95% CI	*p*	AOR 95% CI	*p*
*Dyadic parental risk factors*								
Prior perpetration	**0.50 [0.42,0.61]**	**<0.001**	**2.26 [1.57,3.24]**	**<0.001**	0.66 [0.41,1.08]	0.096	0.87 [0.52,1.46]	0.593
Substance use	**0.83 [0.69,0.99]**	**0.035**	**2.72 [1.82,4.06]**	**<0.001**	**0.19 [0.10,0.36]**	**<0.001**	1.22 [0.63,2.35]	0.549
Mental health problems	0.76 [0.52,1.11]	0.159	1.12 [0.53,2.36]	0.776	1.84 [0.96,3.53]	0.069	0.97 [0.31,3.05]	0.964
Disability and medical conditions	0.87 [0.61,1.22]	0.413	0.69 [0.38,1.26]	0.232	**1.97 [1.03,3.75]**	**0.039**	1.09 [0.32,3.71]	0.891
Inadequate housing	**0.63 [0.48,0.81]**	**<0.001**	**2.29 [1.44,3.64]**	**<0.001**	0.56 [0.25,1.26]	0.162	0.75 [0.39,1.43]	0.377
Economic insecurity	1.16 [0.95,1.42]	0.153	0.95 [0.65,1.40]	0.799	0.81 [0.46,1.42]	0.454	0.62 [0.36,1.05]	0.072
Intimate partner violence	0.88 [0.73,1.06]	0.171	**1.93 [1.34,2.78]**	**<0.001**	0.74 [0.41,1.31]	0.295	**6.71 [3.64,12.37]**	**<0.001**
*Sociodemographic factors*								
Father’s age	0.99 [0.98,1.01]	0.297	1.01 [0.99,1.05]	0.471	1.03 [1.00,1.07]	0.086	0.98 [0.94,1.02]	0.370
Mother’s age	0.99 [0.98,1.01]	0.524	1.02 [0.98,1.05]	0.413	0.97 [0.93,1.01]	0.156	1.05 [1.00,1.11]	0.065
Couple’s race and ethnicity (ref.: Non-Hispanic/Latinx White)								
Non-Hispanic/Latinx Black	**1.41 [1.12,1.78]**	**0.004**	**0.60 [0.39,0.92]**	**0.020**	**0.41 [0.19,0.87]**	**0.021**	2.11 [0.87,5.11]	0.097
Hispanic/Latinx	0.88 [0.60,1.18]	0.397	0.62 [0.37,1.03]	0.062	1.64 [0.87,3.09]	0.124	0.59 [0.15,2.34]	0.448
Other	1.06 [0.85,1.33]	0.583	1.07 [0.69,1.65]	0.778	0.99 [0.54,1.81]	0.963	**2.11 [1.17,3.81]**	**0.014**
Child age	**0.95 [0.92,0.98]**	**<0.001**	0.99 [0.94,1.03]	0.516	**1.21 [1.15,1.27]**	**<0.001**	**1.08 [1.01,1.15]**	**0.022**
Child sex (female)	0.96 [0.82,1.13]	0.597	0.84 [0.62,1.13]	0.248	**4.24 [2.56,7.02]**	**<0.001**	0.90 [0.54,1.49]	0.674

Notes. AOR = Adjusted Odds Ratio. CI = Confidence Interval. Bolded indicates significant adjusted odds ratios. All models included state dummy variables.

## Data Availability

National Child Abuse and Neglect Data System (NCANDS) Child File used for the current study may be requested from the National Data Archive on Child Abuse and Neglect (NDACAN) at Cornell University where the data are stored: https://www.ndacan.acf.hhs.gov/datasets/datasets-list-ncands-child-file.cfm (accessed on 10 February 2022).

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
