# Peer review of "Father–Mother Co-Involvement in Child Maltreatment: Associations of Prior Perpetration, Parental Substance Use, Parental Medical Conditions, Inadequate Housing, and Intimate Partner Violence with Different Maltreatment Types"

_children, 2023, doi:10.3390/children10040707_

Round 1

Reviewer 1 Report

The paper presents an empirical study on father-mother co-involvement in child maltreatment.

The manuscript is well written and understandable.

The research topic is of high relevance for both research and practice.

Important strengths include the multifaceted view on different risk factors and thus explanatory mechanisms.

The paper could nicely fit in the Special Issue “Child Abuse and Neglect”.

The title could be improved. For instance, it could be specified which risk factors in detail are among the key findings.

The conceptual implications could be elaborated in more depth. How are current theoretical models on child abuse advanced? Thereby, the conceptual novelty / contribution will be clearer.

The research topic is highly relevant for child development. The practical implications could be illustrated in more detailed examples.

Which role did gender play? Were there differences between boys vs. girls regarding the associations observed?

Also there may be more or less critical developmental phases. Were there differences according to different age groups of the children?

The social context may be a key risk factor. Did it play a role how many children there were in the household and if there were grandparents living nearby or not?

Author Response

Comments from Reviewer 1:

  1. Comment: The paper presents an empirical study on father-mother co-involvement in child maltreatment. The manuscript is well written and understandable. The research topic is of high relevance for both research and practice. Important strengths include the multifaceted view on different risk factors and thus explanatory mechanisms. The paper could nicely fit in the Special Issue “Child Abuse and Neglect”.

Response: Thank you for the very positive review of this manuscript. We appreciate the reviewer’s comments.

  1. Comment: The title could be improved. For instance, it could be specified which risk factors in detail are among the key findings.

Response: This is a helpful suggestion. Given that prior perpetration, substance use, inadequate housing, and intimate partner violence emerged as key risk factors across different child maltreatment types, we have now modified the manuscript titled to now be: “Father-Mother Co-Involvement in Child Maltreatment: Associations of Prior Perpetration, Substance Use, Inadequate Housing, and Intimate Partner Violence with Different Maltreatment Types”

  1. Comment: The conceptual implications could be elaborated in more depth. How are current theoretical models on child abuse advanced? Thereby, the conceptual novelty / contribution will be clearer.

Response: Thank you for this comment and suggestion. We have now elaborated on the conceptual implications of using the family systems theory to advance current theoretical models on child abuse and neglect. We also made sure to note the conceptual novelty and contribution of the current study. Specifically, on page 10, we added the following sentences: “This study contributes to the limited literature on mother-father co-involvement in child maltreatment, especially by employing the family systems theory. The family systems theory argues for viewing families as integrated whole systems, with mothers and fathers serving as interrelated and interdependent subsystems of larger family systems. Not only are mothers and fathers data used jointly, but also the entire family serves as the unit of analysis in family systems informed studies. Given the generally atheoretical nature of prior studies on the topic of mother-father co-involvement in child maltreatment,2,8,9 application of the family systems theory is a key contribution of this study, challenging current theoretical models on child abuse to consider the joint roles of mothers and fathers and risk factors that occur at the parental dyad or family level.”

We also noted other ways in which the family system theory could be applied in this area of research and thus move the field forward. Specially, on page 13, under the “Limitations and Future Research Directions” we added the following: “A future application of the family system theory in this area of research would be to examine how mothers and fathers establish alliance, communicate, and execute child maltreatment jointly. Because of limited data availability, we were unable to examine such dyadic and interpersonal processes that culminate in mother-father co-involvement in child maltreatment. Furthermore, although not feasible for the current study, future research could extend the family systems theory to examine the consequences of mother-father co-involvement in child maltreatment on parent-child dyads (e.g., betrayal, loss of family ties, and mistrust children may feel as a result).”

  1. Comment: The research topic is highly relevant for child development. The practical implications could be illustrated in more detailed examples.

Response: Thank you for this helpful feedback and suggestion. We have now illustrated in more detail examples of practice implications of our study results. Specifically, on page 13 we noted the importance of child welfare practitioners conducting comprehensive assessments of family risk factors and providing services that ensure the wellbeing of children and families across multiple domains. Specifically, the following sentences were added: “Child welfare practitioners would do well to engage in comprehensive assessments of risk factors at the dyadic parental and family levels, as well as engage in service referrals, resource provisions, and treatments that ensure the wellbeing of families and their children across multiple domains (e.g., housing, physical and mental health, interparental and parent-child relationships, criminal justice involvement).”

We also added on the same page, family group decision making (FGDM) as an example of culturally responsive child welfare practice. Specifically, the following content was added: “For example, family group decision making (FGDM) is a child welfare practice that reflects the cultural values of kinship and community often seen in Black and Indigenous groups.60 FGDM brings together both immediate and extended family members, trained facilitators, community members, and agency personnel to develop child safety and wellbeing plans.60 Through a communal process, family members are encouraged to define their own groups, have a voice in the decision-making process, and remind other stakeholders that children belong with their families and within their kinship networks.60

  1. Comment: Which role did gender play? Were there differences between boys vs. girls regarding the associations observed?

Response: Thank you for this question. In response to the reviewer’s question, we ran subanalyses by child sex which are now provided as Supplemental Material 1. There were both similarities and differences across child sex. We report notable differences here. With regards to physical abuse, substance use predicted lower odds of physical abuse of girls but not boys, and inadequate housing predicted lower odds of physical abuse of boys but not girls. Regarding neglect, the presence of a prior perpetrator in the family and intimate partner violence predicted higher odds of neglect of boys but not girls. Concerning emotional abuse, substance use predicted lower odds of emotional abuse of boys but not girls. Related to sexual abuse, substance use predicted lower odds of sexual abuse of girls only, whereas mental health problems predicted higher odds of sexual abuse of boys only. A number of different patterns were found across sociodemographic factors as well. For details, please see Supplemental Material 1.

  1. Comment: Also there may be more or less critical developmental phases. Were there differences according to different age groups of the children?

 Response: In response to the reviewer’s question, we also ran subanalyses by child age—infancy and toddlerhood (0-2 years), early childhood (3-5 years), school age (6-12 years), and adolescence (13-17 years)—which are now provided as Supplemental Material 2. Given very small cell sizes for sexual abuse and emotional abuse, we decided to focus our subanalyses on physical abuse and neglect. For physical abuse, presence of a prior perpetrator in the family predicted lower odds of physical abuse for all age groups except children in early childhood. Substance use was linked with lower odds of physical abuse of school age children and adolescents only. Inadequate housing and intimate partner violence were linked with lower odds of physical abuse of infants/toddlers only. Concerning neglect, presence of a prior perpetrator in the family, substance use, and inadequate housing predicted higher odds of neglect of infants only. Intimate partner violence emerged as a predictor linked with higher odds of neglect of school age children only. Most of the sociodemographic factors did not serve as significant predictors in these models. For details, please see Supplemental Material 2.

  1. Comment: The social context may be a key risk factor. Did it play a role how many children there were in the household and if there were grandparents living nearby or not?

Response: Thank you for this comment and question. The NCANDS dataset we used for our current study did not have variables that captured the number of children in the household, as well as whether grandparents were living nearby. That said, we agree with the reviewer about the role of social context and the risk it may pose to mother-father co-involvement in child maltreatment. As such, we noted inclusions of such variables as part of future research directions in this area of research. Specifically, on page 12 of the revised manuscript, we added the following content: “Relatedly, given data limitations, we were unable to include measures that capture social context as a potential risk factor. For example, future research would do well to examine the total number of children in the household and how far family members, such as grandparents and other relatives, live from the target families as social context (e.g., crowded house, lack of accessible social and family support) risk factors associated with mother-father co-involvement in child maltreatment.”

Reviewer 2 Report

The study is focused on ab under-examined area in child maltreatment studies, i.e. mother and father co-involvement in different types of child maltreatment (physical abuse, neglect, sexual abuse, and emotional abuse). The analysis is based on national child welfare administrative data that include socio-demographic information about the parents and the forms of abuse. The preparation of the sample is precisely described as well as the methods of the exploratory and multivariate analyses used to test the effect of selected variables related to mother-father co-involvement in child maltreatment.

The topic of the study, the findings from previous studies, the methods and the results from the statistical analysis are very clearly described. There are also interesting insights related neglected aspects of child maltreatment such as the “normalized” and culturally “acceptable” parenting practices that can overlap with some forms of abuse (e.g. the perceptions of strict child discipline and physical abuse in the Black community). Overall, this is very written paper with solid empirical argumentation of the conclusions and I have a few suggestions about it. My furst suggestions to the authors are to explain in more detail how family systems approach applies in the field of mother-father co-involved physical abuse. Another suggestion is to include a short comment (in the Limitations of the study) on the absence of additional characteristics of the parents/household and the children such as employment status of parents, the composition of family (both biological parents, recomposed family, etc.), size of the household, number of siblings, history of previous maltreatments and violence, participation in parenting education programs, child’ disability status, etc. These factors can be associated also with the risk of child maltreatment but they are not included in the models due to absence of information. My third suggestion is to added a comment (in the Conclusion) on the contribution of the study for the development of child maltreatment and prevention programs and for the improvement of the social services that should take into account the cultural practices of child discipline and child upbringing in different communities that may overlap with some forms of maltreatment such as the physical punishment.

Author Response

Comments from Reviewer 2:

  1. Comment: The study is focused on an under-examined area in child maltreatment studies, i.e. mother and father co-involvement in different types of child maltreatment (physical abuse, neglect, sexual abuse, and emotional abuse). The analysis is based on national child welfare administrative data that include socio-demographic information about the parents and the forms of abuse. The preparation of the sample is precisely described as well as the methods of the exploratory and multivariate analyses used to test the effect of selected variables related to mother-father co-involvement in child maltreatment. The topic of the study, the findings from previous studies, the methods and the results from the statistical analysis are very clearly described. There are also interesting insights related neglected aspects of child maltreatment such as the “normalized” and culturally “acceptable” parenting practices that can overlap with some forms of abuse (e.g. the perceptions of strict child discipline and physical abuse in the Black community). Overall, this is very written paper with solid empirical argumentation of the conclusions and I have a few suggestions about it. 

Response: Thank you for the positive review of the manuscript. We very much appreciate the reviewer’s comments and feedback.

  1. Comment: My first suggestions to the authors are to explain in more detail how family systems approach applies in the field of mother-father co-involved physical abuse.

Response: Thank you for this very helpful suggestion. We have now added a new paragraph in the Introduction section of the revised manuscript, detailing how the family systems approach applies to the field of mother-father co-involved child maltreatment. Specifically on page 2 of the revised manuscript, we added the following content: “The family systems theory also applies to the field of mother-father co-involved child maltreatment in that it focuses on the interactions that occur between mothers and fathers and culminate in both members committing child abuse and neglect as joint perpetrators. From a family systems perspective, the entire family—instead of the individual mother, father, or child—is the unit of analysis, as well as focus of intervention and treatment. Because problems like mother-father co-involvement in child maltreatment are seen as a result of what happens in mothers’ and fathers’ interactions with one another, all behaviors—both from mothers and fathers—ought to be examined within the context of the larger family system. Furthermore, given that interactions and interdependence are emphasized within the family system, mother-father co-involvement of child maltreatment suggests alliance, communication, and coordination between mothers and fathers in co-conspiring and co-perpetrating physical abuse, neglect, emotional abuse, or sexual abuse. Mother-father co-involvement in child maltreatment also suggests betrayal of family relations and roles, especially within the parent-child dyad. Such betrayal may leave children feeling as if they are unable to trust or comfortably interact with both their mothers and fathers.”

  1. Comment: Another suggestion is to include a short comment (in the Limitations of the study) on the absence of additional characteristics of the parents/household and the children such as employment status of parents, the composition of family (both biological parents, recomposed family, etc.), size of the household, number of siblings, history of previous maltreatments and violence, participation in parenting education programs, child’ disability status, etc. These factors can be associated also with the risk of child maltreatment but they are not included in the models due to absence of information.

Response: Thank you for this suggestion. Reviewer 1 raised a similar point (i.e., Reviewer 1’s Comment #7 above). In responding to both reviewers’ suggestion, we have now added the following content on page 12 of the revised manuscript: “Relatedly, given data limitations, we were unable to include measures that capture social context as a potential risk factor. For example, future research would do well to examine the total number of children in the household and how far family members, such as grandparents and other relatives, live from the target families as social context (e.g., crowded house, lack of accessible social and family support) risk factors associated with mother-father co-involvement in child maltreatment. Furthermore, absence of relevant information in the data prevented us from including additional characteristics of the children and parents (e.g., parental employment status, family composition, participation in parenting education programs, children’s disability status) as potential risk factors in our analytic models. Overall, future research in this area should aim to include and examine such social context and family characteristics in their associations with mother-father co-involved child maltreatment.”

  1. Comment: My third suggestion is to added a comment (in the Conclusion) on the contribution of the study for the development of child maltreatment and prevention programs and for the improvement of the social services that should take into account the cultural practices of child discipline and child upbringing in different communities that may overlap with some forms of maltreatment such as the physical punishment.

Response: We appreciate the reviewer’s suggestion. In response, we have now added the following content in the Conclusion section on page 13 of the revised manuscript: “Furthermore, child maltreatment prevention programs and social services should take into account the cultural practices of child discipline and upbringing (i.e., physical punishment) in different racial and ethnic communities. Such programs and services should not only recognize overlaps between physical punishment and some forms of child maltreatment like physical abuse, but also aim to understand where physical punishment might be stemming from for parents of color, including Black parents (e.g., historical injustices such as slavery, fear that their children will get in trouble with law enforcement, high levels of socioeconomic disadvantages and ensuing economic stress).”

Reviewer 3 Report

The paper is wonderfully written, the references are relevant and up-to-date, and the methodology is sound considering the available data. The authors state study limitations, future directions and policy implications.

I have just some minor suggestions:

"Although non-biological  248 caregivers  also  engage  in  child  maltreatment, 1 the  current  study  focused  on  biological  249 mothers and fathers of the same child to streamline our analyses."

The authors refer to biological parents in the Method and Limitations section. It would be helpful to explain to the reader more specifically why they opted to analyze biological parents only (e.g., "streamline our analyses").

After the authors' analysis of missing data, 6,996 families were used in the study. They mention that results are not generalizable, yet is there any educated guess possible as to how generalizable they might be considering states included? For example, are there any state characteristics that could be linked to a higher/lower prevalence of the outcomes studied? Second, is it possible to examine or report (not necessarily extensively - perhaps in a sentence or two) whether predictors of the outcomes are largely similar across states included in the sample?

All in all, a great paper and congrats to the authors. 

Author Response

(The authors gave the same response as above.)
